# NMR Lipid Profile of Milk from Alpine Goats with Supplemented Hempseed and Linseed Diets

**DOI:** 10.3390/molecules25071491

**Published:** 2020-03-25

**Authors:** Antonella Caterina Boccia, Erica Cusano, Paola Scano, Roberto Consonni

**Affiliations:** 1CNR, Istituto di Scienze e Tecnologie Chimiche “Giulio Natta”—SCITEC, via A. Corti 12, 20133 Milano, Italy; erica.cusano@cnr.it (E.C.); roberto.consonni@scitec.cnr.it (R.C.); 2Department of Life and Environmental Sciences, University of Cagliari, via Ospedale 72, 09124 Cagliari, Italy; scano@unica.it

**Keywords:** hempseed, linseed, goat’s milk, NMR spectroscopy, supplemented diets, fatty acids, rumenic acid

## Abstract

The supplementation of goat diets with natural products to obtain milk with nutraceutical components is a common practice. In these last years, the influence of supplementation of specifically designed diets has been studied with different analytical tools in order to explore possible beneficial effects in human consumption of animal milk and milk-derived products. In this study, the lipid fraction of milk from Alpine goats undergoing different dietary regimens was studied by ^1^H-NMR spectroscopy. Alpine goats were fed with linseed or hempseed supplements, and after 14 weeks of treatment, milk was collected and analyzed. Results showed that feeding diets supplemented with seeds positively affected the fatty acid composition with a pronounced increase in unsaturated fatty acids for both diets compared to a control diet. Specifically, linolenic acid content was more than doubled for linseed diet compared with the hempseed and control groups, while linoleic acid greatly increased only upon hempseed supplementation. However, a number of conjugated linoleic acid (CLA) isomers and higher levels of fatty acids with *trans* configuration were found in supplemented diets, particularly in the linseed diet.

## 1. Introduction

There is a growing interest in the production of goat milk and derived products, going beyond nutritional contribution and related to their well-recognized role in human health as they are rich in medium-chain triacylglycerols (MCT) [1], which are considered to be a quick energy source and not stored as body fat. Many other health benefits can be associated with the consumption of these products because they represent a source of high-quality proteins and high content of short-chain fatty acids (C6:0, C8:0, and C10:0), therefore offering an alternative for consumers with specific sensitivity to bovine milk [2].

The intensification of goat milk production led to milk-derived products enriched in saturated fatty acids (SFA), but with poor content of linoleic acid (C18:2 *n*-6), linolenic acid (C18:3 *n*-3), and generally, polyunsaturated fatty acids (PUFA) [3]. Many factors such as breed, milking season, and feeding affect milk composition, and feeding appears to be one of the most important factors because it directly influences the quality and properties of milk and represents a natural way for modulating the fatty acid (FA) composition. Several different nutritional strategies have been explored for optimizing the FA composition of milk by supplementing goats’ diets with natural products. Recent studies established that the introduction of flaxseed in an animal’s diet positively impacted the *n-3* FA content of milk from cows, [4,5,6], goats [7,8,9,10], and sheep [11,12]. In contrast to linseed, hempseed supplementation offers a well-balanced linoleic acid/linolenic acid ratio of about 3:1 and contains both stearidonic acid (C18:4 *n*-3) and γ-linolenic acid (C18:3 *n*-6) [13]. The supplementation of ruminants’ diet with hempseed and hempseed oil has been tested and a higher milk fat content with larger proportions of linoleic acid, rumenic acid (C18:2 *cis*-9, *trans*-11), and linolenic acid were observed in both sheep [14] and goats [15].

The administered diets were formulated with the aim of increasing the proportion of unsaturated fatty acids (UFA) in milk because of their potential benefits in human health, and increasing the nutritional quality of milk [16,17,18]. Contemporarily, a positive effect is expected on the human health due to the beneficial effects of conjugated linoleic acid (CLA) in cancer and atherosclerosis prevention, immunomodulation, and a few cardiovascular diseases [19]. Therefore, the production of naturally enriched goat milk with beneficial FA content is desirable for human consumption because it is considered as a functional food with high added value.

In previous studies, the lipid profile of milk and dairy products has been studied by high-resolution NMR spectroscopy for differentiating the animal provenience [20,21,22,23], while other studies reported the presence of different CLA isomers [24,25,26].

This study aimed to evaluate the lipid profiles of milk, derived from goat diets supplemented with natural seeds, by using solution NMR spectroscopy. To this goal, the lipid fraction of milk from Alpine goats fed with diets enriched with linseed or hempseed was analyzed. An exhaustive characterization of milk lipid composition and functional FA was done and data were evaluated in terms of dietary effects.

## 2. Materials and Methods

### 2.1. Sampling

Goat milk samples were collected under the experimental design reported [13]. Briefly, in the period from February to September, 18 selected Alpine goats were divided into three groups and randomly assigned to three different dietary treatments (diets in Appendix A) identified as follows: (1) control diet (group C), (2) diet supplemented with linseed (group L), and (3) diet supplemented with hempseed (group H). Goats gave birth in February and after one month, linseed and hempseed were introduced in the diet for about 14 weeks. Milk samples were collected and pooled within the same group to average composition on each group.

### 2.2. NMR Sample Preparation and Analysis

To separate the fat fraction, milk was centrifuged at 12,100 rcf (relative centrifugal force) for 10 min at 4 °C, then cooled and held at 4 °C. The samples for NMR analysis were obtained by dissolving 1.6 g of fat milk in 3 mL of deuterated chloroform, (d-chloroform 99.8% from Cortecnet, Les Ulis, France) vortexing for 10 min at room temperature, and then centrifuging at 12,100 rcf for 10 min. Successively, 600 µL of the recovered organic phase was placed into an NMR tube. ^1^H- and 2D-NMR experiments were performed on a Bruker Avance 600 DMX spectrometer operating at 14.1 T, equipped with a 5 mm probe and gradient unit on z, at 298 K (Bruker Biospin GmbH, Rheinstetten, Karlsruhe, Germany). Acquisition parameters for ^1^H-NMR experiments were as follows: 90° pulse of 7.85 µs at 2.2 dB, a relaxation delay of 10 s, a spectral width of 7184 Hz, and 256 transients. A resolution enhancement function with an exponential line broadening of 0.5 Hz was applied before Fourier transformation, together with a zero-filling to 32 K points (TopSpin 4.0.6 software, Bruker Biospin GmbH, Rheinstetten, Karlsruhe, Germany). The resolution functions are typically applied before the Fourier transformation to improve the S/N ratio, attenuating the excess noise of poor S/N ratio spectra. The exponential function was applied and the exponential was set to 0.5 Hz. Spectra were referenced to the residual solvent signal at 7.25 ppm, using TMS (tetramethylsilane) as a reference at 0 ppm. Each proton spectrum was recorded in triplicates. 2D ^1^H-^1^H TOCSY experiments (total correlation spectroscopy) were acquired with 256 experiments over 2 K data points and 256 scans each, with a mixing time of 0.08 s and a relaxation delay of 1.2 s. 2D ^1^H ^13^C g-HSQC (gradient-enhanced heteronuclear single-quantum coherence) experiments were acquired with the data matrix 2 K × 512, by acquiring 64 scans, with a relaxation delay of 1.5 s and spectral widths of 7184 Hz and 24,200 Hz for ^1^H and ^13^C dimensions, respectively. The NMR spectra were processed using the TopSpin 4.0.6 software. 1D ^13^C-NMR experiments were acquired on a Bruker 400 MHz spectrometer with a spectral width of 36 KHz, a relaxation delay of 16 s, and 90° pulse of 11.0 µs at −5.0 dB. The spectra were then referenced to the residual solvent signal at 77.1 ppm. After calibrating the ^1^H and ^13^C spectra of the milk fat from C group, the spectra from groups H and L were scaled with respect to group C, as little differences on the resonance position ascribed to small shifts due to changes in pH values and concentration gradients were compensated. 

## 3. Results

The effects of the supplement diets on goat milk were evaluated by comparing the quantitative ^1^H spectra of the lipid fraction for the three groups (C = control, L = linseed, and H = hempseed) after 14 weeks of supplemented diets. Figure 1 illustrates the typical ^1^H-NMR spectrum of the milk lipid fraction (full ^1^H-NMR spectrum in Appendix A). The resonances of triacylglycerols (TAG) dominated the spectrum as expected. Signals of glycerol were located respectively at 5.23 ppm (*sn*-2, CH TAG), 4.27 ppm (*sn-*3 and *sn*-1 Hb protons, CH_2_ TAG), and 4.11 ppm (*sn*-1 and *sn*-3 Ha protons, CH_2_ TAG). The olefinic protons of all unsaturated chains were detected in the range of 5.30–5.34 ppm; and the α-methylenic protons of all acyl chains of FA appeared at 2.27 ppm as a well-resolved triplet, while the β-methylenic protons appeared at 1.58 ppm as a multiplet. Finally, the methylenic protons for all types of FA occurred at 1.25 ppm, while protons of the corresponding methyl groups occurred at 0.85 ppm.

The assignment of the minor components of fat milk reported in Table 1, along with the experimental measured chemical shifts and some of the other principal resonances, was performed according to the available literature data, [20,24,27], ^13^C-NMR data (full spectrum in Appendix A), and 2D spectroscopy (TOCSY full spectrum in Appendix A and HSQC experiments in Appendix A).

The minor components identified were CLA, caproleic acid (C10:1 *cis*-9), diacylglycerols (DAGs), linoleic acid, and linolenic acid. CLA is a class of PUFA constituted by a mixture of geometric (*cis*-*cis*, *trans*-*trans*, *cis*-*trans*, and *trans*-*cis*) and positional (6–8 to 13–15) isomers [28]. From the analysis of the unsaturated region reported in Figure 2, CLA was prevalently present as the *cis-*9, *trans*-11 isomer (also named rumenic acid), confirmed by the characteristic pattern of signals and *^3^J* coupling constant, while in minor amount as the *trans-9*, *trans-11* isomer. In addition, caproleic acid was detected due to its specific pattern of signals, specifically as a multiplet at 5.75 ppm and as a double quartet at 4.94 and 4.88 ppm.

Glycerol protons of 1,2-DAG appeared as multiplets at 5.06 ppm and 3.67 ppm, respectively; 1,3-DAG protons were identified on the basis of the typical signal at 3.98 ppm. A careful inspection of the TOCSY spectrum in the olefinic region, shown in Figure 3, allowed to identify, for the first time in milk, particularly fat milk from linseed diet, the presence of both CLA *cis*-9, *cis*-11 and *cis*-10, *cis*-12 isomers by NMR spectroscopy.

Finally, the presence of linolenic acid and linoleic acid (Figure 4) was confirmed on the basis of the specific pattern of the *bis*-allylic methylene signals assigned to the triplets occurring at 2.77 ppm (t) for linolenic acid and at 2.73 ppm for linoleic acid. Beside these resonances, two smaller peaks were observed (Figure 4) at 2.67 and 2.80 ppm, that were tentatively assigned to *bis*-allylic methylene protons, probably in *trans-cis* configurations, for both linoleic acid and linolenic acid, respectively. The quantitative analysis, as reported in Table 2, was performed by normalizing the area of NMR signals in the proton spectrum to that of the α-methylene signal of all FA at δ = 2.27 ppm and scaling the integral values for all the other components of the mixture. The calculated linoleic acid and linolenic acid molar ratios were in agreement with those evaluated with the method of Brescia [23]. The percentage of UFA was obtained as follows: UFA = (I7 + I8)/2. Therefore, the monounsaturated fatty acids (MUFA) were determined as follows: MUFA = UFA − (linoleic + linolenic) and the SFA were determined as: SFA = all FA − UFA. The quantitative evaluation of the lipid fraction of fat milk samples is listed in Table 2. Here, a decrease of SFA in favor of UFA can be observed in goat milk under L and H diets compared to the control. MUFA and CLA *cis-9*, *trans-11* increased together with the expected increase of PUFA (linoleic acid and linolenic acid). While a little increase in linoleic acid was observed, linolenic acid markedly increased in the linseed diet, together with UFA with a *trans* double bond. DAGs and caproleic acid were found at lower levels in H and L diets.

The effects on fat milk profile due to the supplemented diets were also evaluated by comparing the ^13^C-NMR spectra for each group. The olefinic carbon region in the spectra clearly showed an increase in the resonances for both H and L diets compared to the control, thus indicating a large increase in the diversity of UFA (Figure 5).

Moreover, another interesting spectral region of the ^13^C-NMR spectra was in the methylenic carbon region at 33–31 ppm range (Figure 6). Here, strengthening the indication from the analysis of the ^1^H-NMR data, the higher levels of FA with *trans* configuration in the L and H diets could be easily seen. The presence of 1,2-DAG and 1,3-DAG in fat milk samples was confirmed with the ^13^C-NMR experiments, specifically in the 60–72 ppm spectral region.

## 4. Discussion

As evidenced by the NMR analysis, supplementation of Alpine goat diet with linseed and hempseed positively affected the milk fat profile by increasing the UFA content. UFA, and in general long-chain FA, are not synthesized by tissues in ruminants, therefore their concentration in milk is closely related to the quantities absorbed in the intestine, hence the quantities leaving the rumen may be increased by dietary UFA. Accordingly, increasing UFA diet with oil seeds led to an increase in UFA in milk from 19% in the control group (C) up to 32% and 25% for linseed (L) and hempseed (H) groups, respectively. The linolenic acid content was more than double for the L group compared with the H and C groups (Table 2). Linoleic acid content percentage presented a different trend; it was almost the same for C and L groups, while a small increase in the percentage was observed in the H group compared to the C group.

Moreover, UFA present in the diet undergo bio-hydrogenation in the rumen, producing a number of FA with different lengths and unsaturation degrees. The principal products in milk are *trans*-vaccenic acid (C18:1 *trans*-11) and rumenic acid, the latter is also produced by the *Δ9*-desaturase enzyme in the mammary gland. Accordingly, in both diets rich in UFA, an increase in rumenic acid and *trans*-vaccenic acid was observed, with higher levels in the L diet, although accompanied by the presence of other CLA isomers. Considering the health benefits derived from consumption of these FA, naturally enriched milk fat can be considered as a functional food for human consumption [29,30].

Feeding lipid supplements to dairy ruminants has been widely used for decades by researchers and, to some extent, by farmers to modify milk FA composition. Attempts to change the proportion of one category of FA often induce changes in other FA, which may be considered as a positive or negative effect for consumer health. Indeed, diets that decrease saturated FA and increase PUFA and/or CLA in milk, generally result in higher *trans*-FA proportions, whose effects are still controversial [31]. In agreement with this observation, diets L and H, albeit increasing the milk UFA level, led to a higher content of *trans*-FA. At the same time, the effects resulting from the supplemented diets with hempseed and linseed led to a decrease in the content of caproleic acid in both H and L groups compared to the control. Caproleic acid content in milk from goats fed with linseed was almost half of that observed in the C group, and about 30% less for milk from goats fed with hempseed compared to the C group. Considering that caproleic acid is mainly produced by the *Δ9*-desaturase activity on C10:0 in the mammary gland, the lower levels in both the supplemented diets with respect to the control can be linked to a reduced activity of the *Δ9*-desaturase [15,32]. However, we can also hypothesize that the activity of this enzyme is pushed toward the route of C18 FA, of which L and H diets are particularly rich, and this hypothesis is supported by the higher level of MUFA (mostly oleic acid) found in both diets. Changes in the DAG content can be justified considering that they are derived from milk fat lipolysis, which is the hydrolysis of fat globule TAG into free FA and diacylglycerols. In contrast to cow and goat milk, lipolysis was found to decrease when animals were underfed or received a diet supplemented with plant oils [33]. In agreement with that, our data showed that introducing hempseed or linseed into the diets led to a decrease in 1,2-DAG and 1,3-DAG content in both H and L groups.

## 5. Conclusions

The NMR spectroscopy represents a valid and powerful technique for the characterization of the lipid fraction of goat milk and is able to provide a global view of the effects on milk derived from seed-supplemented diets. Quantitative and qualitative results showed that, although an increase in the levels of beneficial FA (UFA and rumenic acid) was obtained, other compounds such as *trans*-FA and CLA isomers were produced, especially in milk obtained from animals that were fed a linseed diet. In conclusion, oilseed supplementation reduced the SFA/UFA ratio and increased nutritionally desirable FA, making the naturally enriched goat milk, a functional food with high added value that is able to provide human health benefits.

## Figures and Tables

**Figure 1 molecules-25-01491-f001:**
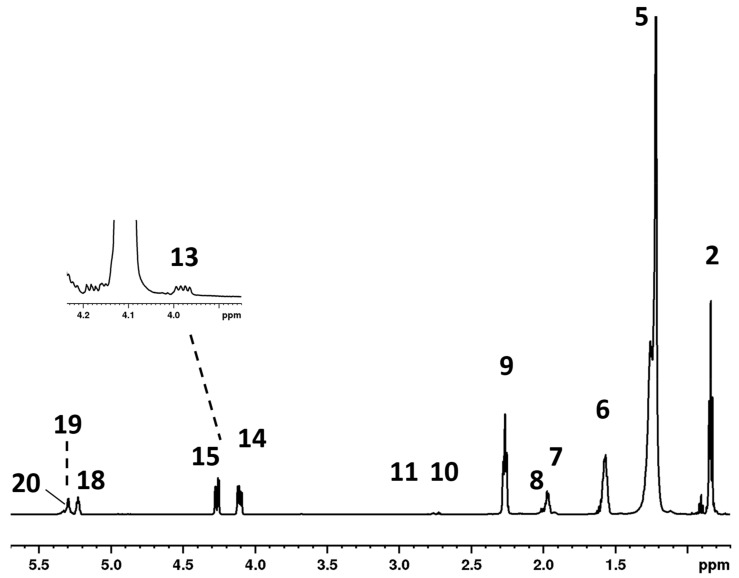
Selected region of ^1^H spectrum of goat milk lipid fraction, recorded on a 600 MHz spectrometer, in CDCl_3_ at 298 K. The inset shows resonances due to 1,3-DAG. Numbering refers to signals assigned in Table 1.

**Figure 2 molecules-25-01491-f002:**
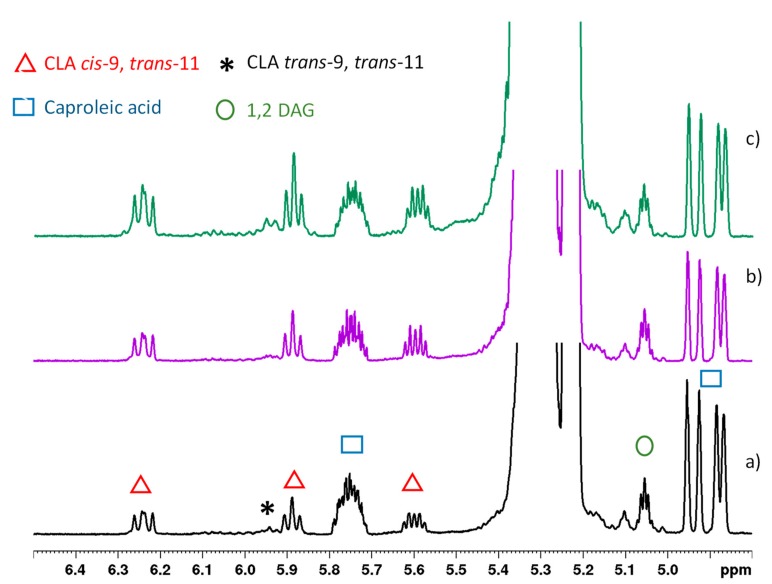
Expanded region of 600 MHz ^1^H spectra of goat milk lipid fraction from (**a**) control, (**b**) hempseed, and (**c**) linseed diets in CDCl_3_ at 298 K. Resonances highlighted with symbols show changes due to diets.

**Figure 3 molecules-25-01491-f003:**
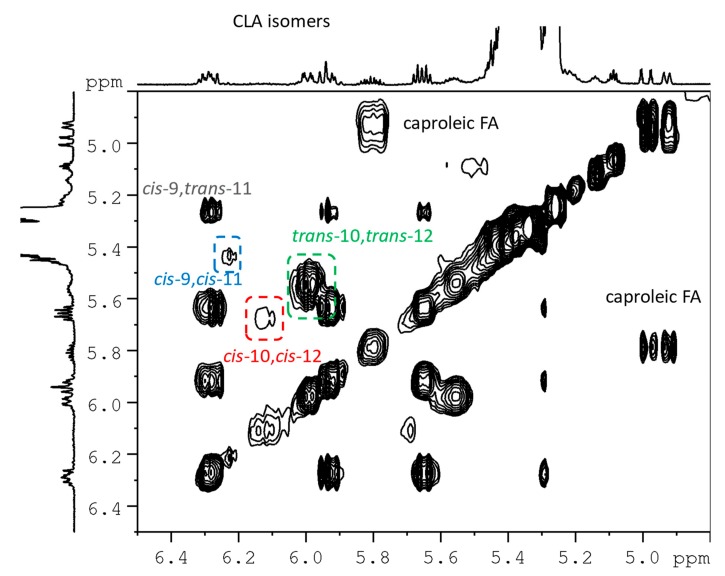
Expanded olefinic region of ^1^H-^1^H TOCSY spectrum of goat milk lipid fraction, recorded on a 600 MHz spectrometer, in CDCl_3_ at 298 K. Highlighted cross-peaks refer to CLA isomers assigned with this technique.

**Figure 4 molecules-25-01491-f004:**
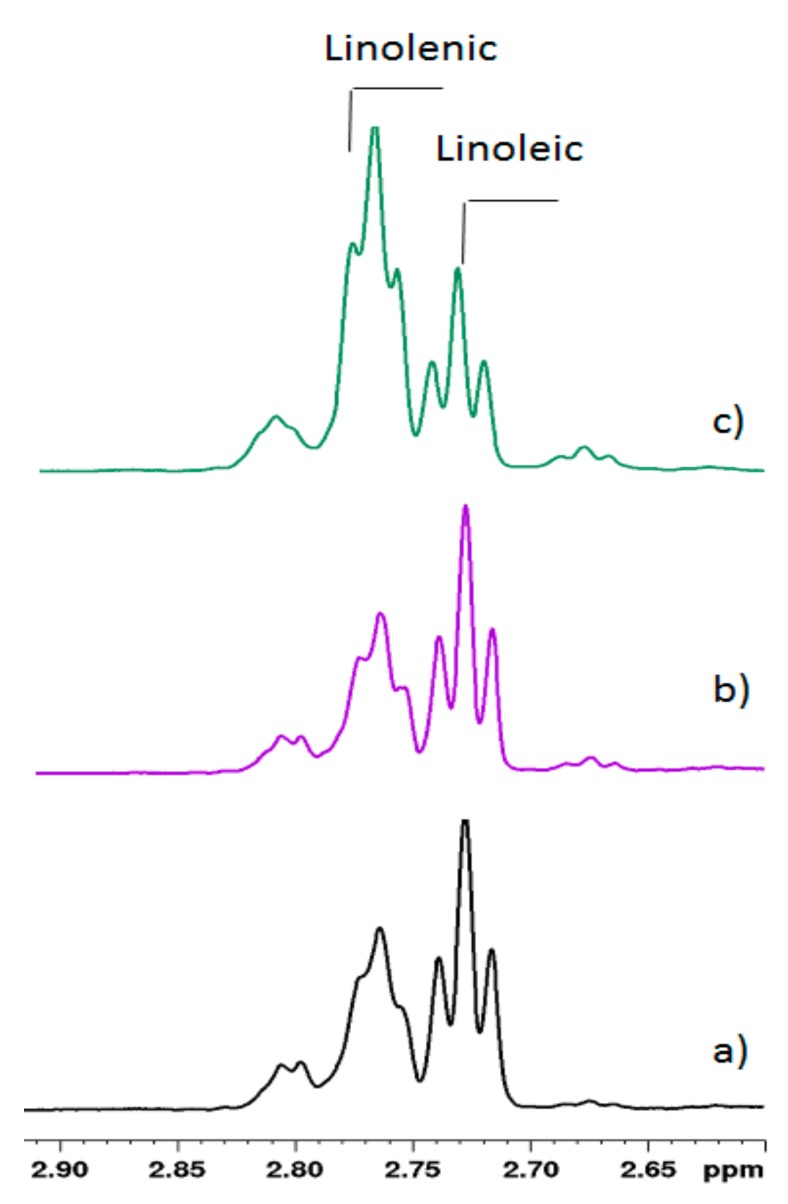
*Bis*-allylic methylene region of 600 MHz ^1^H NMR spectra of goat milk lipid fraction from (**a**) control, (**b**) hempseed, and (**c**) linseed diets in CDCl_3_ at 298 K. Changes in the signal ratios of linoleic acid and linolenic acid depending on the diets are clearly visible.

**Figure 5 molecules-25-01491-f005:**
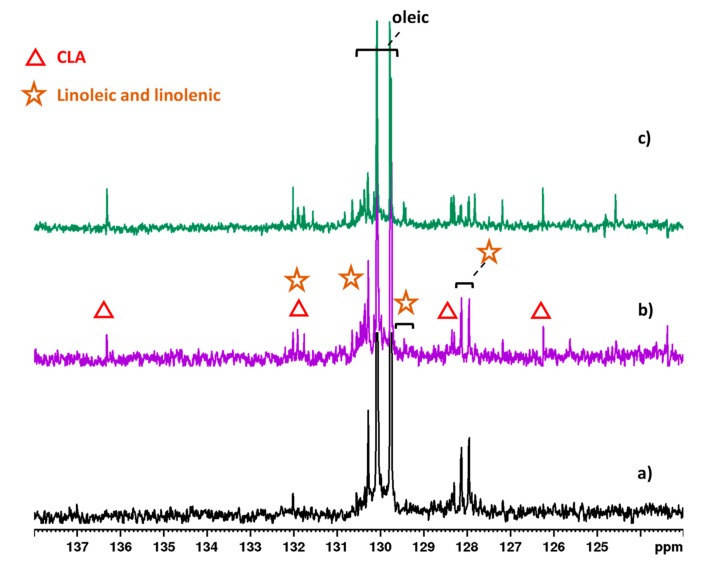
Olefinic region of 400 MHz ^13^C NMR spectra of goat milk lipid fraction from (**a**) control, (**b**) hempseed, and (**c**) linseed diets in CDCl_3_ at 298 K. Resonances highlighted with symbols refer to changes due to diets.

**Figure 6 molecules-25-01491-f006:**
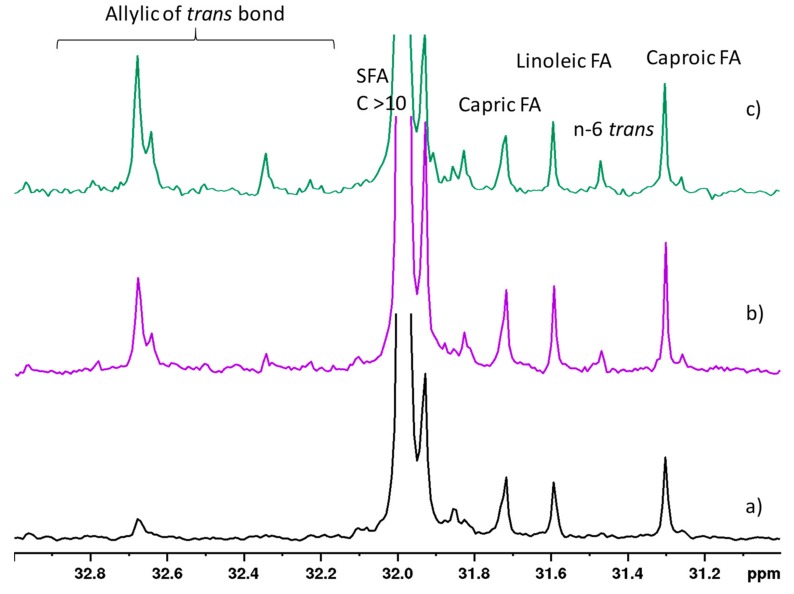
Expanded region of 400 MHz ^13^C-NMR spectra of goat milk lipid fraction from (**a**) control, (**b**) hempseed, and (**c**) linseed diets in CDCl_3_ at 298 K. Resonances highlighted with symbols refer to changes due to diets.

**Table 1 molecules-25-01491-t001:** Peak assignments for ^1^H spectrum of goat milk lipid fraction, recorded on a 600 MHz spectrometer, in CDCl_3_ at 298 K.

Peak	Assignment	Functional Group	δ ^1^H ppm	Multiplicity^1^
**1**	Cholesterol	*-CH_3_*	0.65	s
**2**	FA	-*CH_3_*	0.85	t
**3**	Butyric FA	-*CH_3_*	0.91	t
**4**	All *n*-3 FA	-*CH_3_*	0.95	t
**5**	All FA	-(*CH_2_*)_n_	1.25	m
**6**	All FA	-OOC-CH_2_-*CH_2_*-	1.58	m
**7**	*Trans* UFA	-*CH_2_*-CH=CH-	1.93	m
**8**	*Cis* UFA	*-CH_2_*-CH=CH-	1.98	m
**9**	All FA	-OOC-*CH_2_*-CH_2_-	2.27	t
**10**	Linoleic FA	=CH-*CH_2_*-CH=	2.73	t, or, m
**11**	Linolenic FA	=CH-*CH_2_*-CH=	2.77	t
**12**	Glycerol in 1,2-DAG	OH-*CH_2_*-CH-	3.67	dd
**13**	Glycerol in 1,3-DAG	-*CH_2_*-OOC-	3.98	m
**14**	Glycerol in TAG	-*CH_2_*-OOC-	4.11	dd
**15**	Glycerol in TAG	-*CH_2_*-OOC-	4.27	m
**16**	Caproleic FA	=CH	4.88–4.94	dd
**17**	Glycerol in 1,2-DAG	-*CH*-OOC-	5.06	m
**18**	Glycerol in TAG	-*CH*-OOC-	5.23	m
**19**	*Cis* UFA	-CH=*CH*-	5.30	m
**20**	*Trans* UFA	-CH=*CH*-	5.34	m
**21**	CLA *cis*-9, *trans-*11	=*CH*-	5.60	m
**22**	Caproleic FA	-CH=*CH_2_*	5.75	m
**23**	CLA *cis-9*, *trans-*11	-CH=*CH*-	5.92	t
**24**	CLA *trans-9*, *trans-11*	-CH=*CH*-	5.95	m
**25**	CLA *trans-*10, *trans-*12	-CH=*CH*-	5.99	m
**26**	CLA *cis*-9, *trans-*11	-CH=*CH*-	6.25	dd
**27**	CLA *trans-10*, *cis-12*	-CH=*CH*-	6.24	dd
**28**	CLA *cis-10*, *cis-12*	-CH=*CH*-	6.13	m

Abbreviations: ^1^ s, singlet; m, multiplet; d, doublet; dd, doublet of doublets; t, triplet. FA, fatty acids; UFA, unsaturated fatty acids; TAG, triacylglycerols, DAG, diacylglycerols; CLA, conjugated linoleic acids.

**Table 2 molecules-25-01491-t002:** FA (mol%) composition of milk from three groups of goats under different dietary regimens and fold changes (L or H – C/C)% due to supplements. C = control, L = linseed, H = hempseed.

Compound	Group C	Group L		Group H	
	% mol	% mol	Fold change%	% mol	Fold change%
**All FA (I_9_)^1^**	100	100		100	
**Linoleic acid (I_10_)**	1.86	1.71	-8	1.96	5
**Linolenic acid (I_11_/2)**	0.80	1.80	125	0.85	6
**CLA *cis*-9, *trans*-11 *(*I_26_ ∙ 2*)***	0.40	0.54	35	0.48	20
**CLA *trans, trans* (I_24_)**	0.06	0.10	67	0.07	17
**Caproleic acid (I_16_)**	1.37	0.74	–46	1.01	–26
**MUFA (UFA – PUFA)**	16.47	28.19	71	22.34	36
**UFA (I_7_ + I_8_)/2**	19.13	31.70	66	25.15	31
**PUFA [(I_10_) + (I_11_/2)]**	2.66	3.51	32	2.81	6
**SFA = all FA – UFA**	80.87	68.30	–16	74.85	–7
***Trans*-bond FA (I_20_)**	4.63	10.26	122	7.18	55
**1,2-DAG (I_17_ ∙ 2)**	0.58	0.54	–7	0.32	–45
**1,3-DAG (I_13_)**	0.39	0.27	-31	0.36	-8

^1^ Integral of the peak used for quantitative purpose, numbers refer to assignments in Figure 1 and Table 1.

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
