# Peer review of "NMR Lipid Profile of Milk from Alpine Goats with Supplemented Hempseed and Linseed Diets"

_molecules, 2020, doi:10.3390/molecules25071491_

Round 1

Reviewer 1 Report

Comment on molecules-743500 manuscript:

NMR lipid profile of milk from Alpine goats with 2 supplemented hempseed and linseed diets:

submitted to Molecules

This manuscript reports on a very nice use of 1D NMR spectroscopy to control unsaturated fatty acid presence and relative amount in goat milk resulting from differently supplemented diet. This, the study is a very practical one and is related to our diet enriched in unsaturated fatty acids. In general I like the idea behind this study. However, before the manuscript is accepted I invite the authors to correct some minor problems in the text:

  1. Page 2, line 52: atherosclerosis - check Sperling
  2. Page 3 line 81: check “bidimensional NMR experiments” – what about two-dimensional or 2D spectroscopy.
  3. Page 6, Fig. 4. An integral line with relative intensity of signals is missing,
  4. Page 9, NMR experiment: mono and bidimensial spectra – should be one-dimensial and two (or 2D and 1D NMR spectra)
  5. Resolution enhancement(RE) – and line broadening makes no sense. Line broadening is for improving S/N ratio. In case of RE a name of routine and parameters should be stated. Then a question of signal integration should be explained.
  6. Why TMS was not used as reference? Solvent signal is more sensitive to intermolecular interactions.

Author Response

The Authors are very grateful to the reviewers for their suggestions; the manuscript was modified following them.

Comment on molecules-743500 manuscript:

NMR lipid profile of milk from Alpine goats with 2 supplemented hempseed and linseed diets:

submitted to Molecules

This manuscript reports on a very nice use of 1D NMR spectroscopy to control unsaturated fatty acid presence and relative amount in goat milk resulting from differently supplemented diet. This, the study is a very practical one and is related to our diet enriched in unsaturated fatty acids. In general I like the idea behind this study. However, before the manuscript is accepted I invite the authors to correct some minor problems in the text:

Rev 1

  1. Page 2, line 52: atherosclerosis - check Sperling.

Text has been corrected.

  1. Page 3 line 81: check “bidimensional NMR experiments” – what about two-dimensional or 2D spectroscopy.

Thank you for your suggestion, we have corrected in the text.

  1. Page 6, Fig. 4. An integral line with relative intensity of signals is missing,

Aim of the figure was to give a snapshot of the different intensities of the allylic signals; the integral values scaled as described in the “Results” section are reported in Table 2. Caption of figure has been changed accordingly.

  1. Page 9, NMR experiment: mono and bidimensial spectra – should be one-dimensial and two (or 2D and 1D NMR spectra).

Thank you, we have corrected in the text.

  1. Resolution enhancement(RE) – and line broadening makes no sense. Line broadening is for improving S/N ratio. In case of RE a name of routine and parameters should be stated. Then a question of signal integration should be explained.

Within the text we reported the sentence “A resolution enhancement function with an exponential line broadening of 0.5 Hz was applied before Fourier transformation, together with a zero-filling to 32 K points (TopSpin 4.0.6 software).” The resolution functions are typically applied before the Fourier transformation, to improve the S/N ratio attenuating the excess noise of poor S/N ratio spectra; the exponential function was applied, and the exponential was set to 0,5Hz. The indication of some processing parameters in the text are useful for a correct quantitative analysis of proton spectra, as all the experiments have to be processed by applying the same parameters.

  1. Why TMS was not used as reference? Solvent signal is more sensitive to intermolecular interactions.

We agree with the reviewer comment and we have corrected in the text as follows: The residual solvent signal was calibrated at 7.25 ppm, using TMS as a reference at 0 ppm.

Reviewer 2 Report

Some editing for English language and grammar is needed (for instance, use of the word "nutrimental").

The methods for quantifying FA are described in the Results. Please move these to the Methods. 

Figure and table captions are not very descriptive.

Author Response

The Authors are very grateful to the reviewers for their suggestions; the manuscript was modified following them.

Comment on molecules-743500 manuscript:

Rev 2

Some editing for English language and grammar is needed (for instance, use of the word "nutrimental").

English grammar has been checked and corrected in the text.

       The methods for quantifying FA are described in the Results. Please move these to the Methods.

Text has been modified.

Figure and table captions are not very descriptive.

Thank you for your suggestion, the captions has been modified.